# Efficient Base-Catalyzed Kemp Elimination in an Engineered Ancestral Enzyme

**DOI:** 10.3390/ijms23168934

**Published:** 2022-08-11

**Authors:** Luis I. Gutierrez-Rus, Miguel Alcalde, Valeria A. Risso, Jose M. Sanchez-Ruiz

**Affiliations:** 1Departamento de Quimica Fisica, Facultad de Ciencias, Unidad de Excelencia de Quimica Aplicada a Biomedicina y Medioambiente (UEQ), Universidad de Granada, 18071 Granada, Spain; 2Department of Biocatalysis, Institute of Catalysis and Petrochemistry, CSIC, Cantoblanco, 28049 Madrid, Spain

**Keywords:** enzyme design, de novo enzymes, Kemp elimination, ancestral enzymes, β-lactamases, focused library screening, focused directed evolution

## Abstract

The routine generation of enzymes with completely new active sites is a major unsolved problem in protein engineering. Advances in this field have thus far been modest, perhaps due, at least in part, to the widespread use of modern natural proteins as scaffolds for de novo engineering. Most modern proteins are highly evolved and specialized and, consequently, difficult to repurpose for completely new functionalities. Conceivably, resurrected ancestral proteins with the biophysical properties that promote evolvability, such as high stability and conformational diversity, could provide better scaffolds for de novo enzyme generation. Kemp elimination, a non-natural reaction that provides a simple model of proton abstraction from carbon, has been extensively used as a benchmark in de novo enzyme engineering. Here, we present an engineered ancestral β-lactamase with a new active site that is capable of efficiently catalyzing Kemp elimination. The engineering of our Kemp eliminase involved minimalist design based on a single function-generating mutation, inclusion of an extra polypeptide segment at a position close to the de novo active site, and sharply focused, low-throughput library screening. Nevertheless, its catalytic parameters (k_cat_/K_M_~2·10^5^ M^−1^ s^−1^, k_cat_~635 s^−1^) compare favorably with the average modern natural enzyme and match the best proton-abstraction de novo Kemp eliminases that are reported in the literature. The general implications of our results for de novo enzyme engineering are discussed.

## 1. Introduction

In the early 1960s, Linus Pauling and Emile Zurkenkandl published two papers that played a crucial role in the emergence and subsequent development of the molecular evolution field. In the first paper [1], they introduced the molecular clock hypothesis and the possibility of using sequences of homolog proteins to estimate species divergence times. In the second paper [2], they proposed that plausible approximations to the sequences of ancestral proteins can be derived from suitable analyses of the known sequences of their modern counterparts. Ancestral sequence reconstruction has been amply used in the post-genomic era as a tool to address fundamental problems in molecular evolution [3,4,5]. Furthermore, it has been found that ancestral proteins that are “resurrected” in the lab (i.e., the proteins encoded by the reconstructed sequences) often display interesting and even extreme properties [6,7,8,9,10,11,12,13,14,15,16] that, plausibly, reflect ancestral adaptations to unusual intra- and extra-cellular environments. Resurrected ancestral proteins often display high stability, supporting the frequently hypothesized thermophilic nature of ancient life. In addition, they often show efficient heterologous expression in modern hosts, possibly reflecting their emergence prior to the availability of efficient cellular folding assistance. Plausibly, unlike many modern proteins, they do not rely on having adapted to (i.e., having co-evolved with) the folding assistance machinery of the host to fold efficiently. Finally, in a significant number of ancestral reconstruction studies, the resurrected enzymes were found to be able to catalyze several related reactions. Such promiscuity, to use the accepted term in the field, may be consistent with Jensen’s proposal, many years ago [17], that primordial enzymes were generalists. Alternatively, perhaps the promiscuous resurrected ancestral enzymes corresponded to pre-duplication nodes in the evolution of new functionalities [10].

Beyond specific evolutionary narratives, it is clear that the unusual properties of resurrected proteins may be valuable in biotechnological application scenarios [5,7,12,13,14,18,19,20,21,22]. Promiscuity, for instance, is most likely linked to conformational diversity, i.e., to the fact that the protein exists in solution as an ensemble of more or less related conformations, with different functionalities linked to different subsets of conformations [23]. It is well known that conformational diversity promotes evolvability, i.e., the capability to evolve new functions [24,25,26,27]. The reason for this is that if a few rare conformations are competent for the targeted activity, their population in the ensemble can be enhanced through rational design or, more likely, through standard directed evolution. Enhanced stability, in addition to being a biotechnologically useful property by itself, is known to contribute to evolvability [28], as functionally useful but destabilizing mutations will be accepted in a high-stability protein scaffold, while they will likely compromise proper folding in a moderately stable protein.

Furthermore, and contrary to naïve expectations, enhanced conformational diversity and high stability are not necessarily mutually exclusive features [29]. In an ancestral reconstruction exercise targeting the antibiotic-resistance enzyme β-lactamase, published approximately ten years ago [7], we found that the resurrected proteins corresponding to ancient Precambrian phylogenetic nodes were not only highly stable, with denaturation temperatures about 30 °C above those of their modern mesophilic counterparts, but also promiscuous, being able to degrade a variety of lactam antibiotics, including third-generation antibiotics. For comparison, TEM-1 β-lactamase, a typical modern β-lactamase, is a penicillin specialist that displays very low levels of activity with other lactam antibiotics. Subsequent experimental and computational studies [30,31] confirmed that the substrate promiscuity of the ancestral β-lactamases was, indeed, linked to enhanced conformational diversity. Resurrected Precambrian β-lactamases are, therefore, highly-stable and conformationally diverse to some substantial extent.

Overall, it emerges that resurrected ancestral proteins in general, and our highly stable and conformationally diverse Precambrian β-lactamases in particular, could perhaps provide superior starting points for protein engineering. In order to systematically explore this possibility, we recently started a research program aimed at using our resurrected ancestral β-lactamases as scaffolds for de novo enzyme generation, which is a fundamental unsolved problem in protein engineering [32]. We targeted Kemp elimination (Figure 1), a simple model of a fundamental chemical process—proton abstraction from carbon—and an extensively used benchmark in de novo enzyme engineering. Specifically, we aimed at generating efficient Kemp eliminases with minimal design and screening efforts. In our first step [33], we succeeded in using a single-mutation, minimalist design to generate significant levels of Kemp elimination activity in ancestral β-lactamase scaffolds. In our second step [34], we improved these starting activities on the basis of ultra-low-throughput screening that was computationally focused on the new active site region. 

Here, we report the third step along these lines. The interactions at the original de novo active site have probably been optimized to a substantial extent by our previous efforts. Therefore, we have devised a different approach, based on the extension of the protein via the introduction of an additional polypeptide segment near the new active site. The rationale behind this approach is that the extra segment should display conformational diversity; we may expect that some conformations and sequence patterns generate new interactions that promote catalysis. Overall, the extra polypeptide segment offers new interaction possibilities that can presumably be exploited for de novo activity enhancement. In fact, by screening a small library focused on the additional segment in this work, we reached a catalytic efficiency of k_cat_/K_M_~2·10^5^ M^−1^ s^−1^ and a turnover number of k_cat_~635 s^−1^. These Michaelis–Menten catalytic parameters compare favorably with those for the average modern natural enzyme [35] and match the best proton-abstraction de novo Kemp eliminases reported in the literature [36]. The general implications of these results for de novo enzyme engineering are discussed in Section 3 of this paper.

## 2. Results

### 2.1. Kemp Eliminase Variants Used as Starting Point for This Work

Recently [33], we generated a completely new active site for Kemp elimination—i.e., a site that was distinct from the antibiotic degradation active site—using a minimalist approach based on a single mutation. Specifically, a hydrophobic-to-ionizable mutation generated both a cavity for substrate binding and a catalytic base that was capable of performing proton abstraction. Both the high stability and the conformational diversity of the ancestral β-lactamase scaffolds that were used likely played roles in the success of the minimalist design [33,37]. The function-generating mutation, a tryptophan to aspartate amino acid replacement at position 229, is clearly disruptive and may lead to folding problems if implemented in a β-lactamase of moderate stability. Furthermore, the cavity produced by the mutation cannot exactly match the shape of the Kemp reactant; therefore, substrate binding must rely on local flexibility in the region of the new active site. Indeed, while the minimalist design was successful in a number of resurrected Precambrian β-lactamases, it failed to lead to significant de novo activity levels when implemented in 10 different modern β-lactamases.

The activity generated by W229D was found to be enhanced by a second F290W mutation [33], resulting in a catalytic efficiency of approximately 10^4^ M^−1^ s^−1^ with a turnover number of approximately 10 s^−1^ for Kemp elimination catalysis. Figure 2A shows the 3D structure of the more active de novo Kemp eliminase that we initially obtained [33] through a scan of the function-generating W229D mutation on several resurrected ancestral β-lactamases. That structure includes a bound transition-state analogue (Figure 1B) that indicates the location of the engineered active site.

Subsequently [34], we used computationally focused screening to target the active site region, to further increase the Kemp elimination activity of our best W229D/F290W variant of an ancestral β-lactamase scaffold. The enhancement in catalytic efficiency that was obtained was moderate, but the turnover number was raised by approximately one order of magnitude. The mutations introduced at this stage are highlighted in the structure shown in Figure 2B. Note that in all our Kemp eliminases, the de novo active site is located near the carboxyl terminus and that His-tag is routinely attached to the carboxyl terminus residue to enable protein purification by affinity chromatography. Note also that one of the activity-enhancing mutations shown in Figure 2B actually replaces the first histidine residue of the purification tag with a valine.

### 2.2. Combinatorial Library Design and Screening

In this work, we inserted a polypeptide segment between the carboxyl terminus residue and the His-tag (Figure 3A) of our previous best Kemp eliminase. This insertion retained the valine residue at the first position after the original carboxyl terminal residue, while the rest of the inserted segment included several glycine residues and a serine, following the known sequence design principles for soluble and flexible protein linkers [38]. Overall, the β-lactamase variant used here as starting point for directed evolution was identical to the best Kemp eliminase from our previous study [34], except for the presence of a (Gly)_5_-Ser-Leu-Glu-(His)_6_ segment between the extra valine at the carboxyl terminus and the purification His-tag. This insertion had a small effect on catalysis, as shown in Figure 2C by the Michaelis–Menten plots and catalytic parameters. In fact, as shown in Figure 2D, the effect of the insertion on catalysis was almost negligible when compared with the total activity enhancement that was achieved in this work.

The β-lactamase variant described above was used as background for a library that comprises all combinations of all possible amino acid residues at the three first positions after the carboxyl terminus (Figure 3A). The rationale behind this approach was that, as the carboxyl terminus is close to the de novo active site, some of the library variants may generate interactions that promote catalysis. The combinatorial library spanned 20^3^ = 8000 different amino acid sequences, and approximately 800 variants were screened for Kemp elimination activity, as described in Section 4 of this paper. In this primary screening, most variants displayed significant levels of Kemp elimination activity (Figure 3B), indicating that few of the mutations were disruptive and prevented proper folding. This is consistent with the fact that, in this case, library screening sampled the sequence space associated with a conformationally flexible polypeptide segment that was expected to remain largely exposed to the solvent in most cases. Nevertheless, a few of the variants were found to display substantially enhanced levels of catalysis for the Kemp elimination reaction (Figure 3B). The sequences of these variants were determined by Sanger sequencing; the corresponding combinations of residues at the randomized positions that enhance activity are provided in Table 1. Because catalysis relies on decreasing the activation free energy of the reaction, it appears reasonable to assume that these variants can generate interactions that stabilize the transition state (i.e., the chemical species at the top of the free energy barrier) for Kemp elimination at the de novo active site.

### 2.3. Stability and Catalytic Parameters for the Improved Kemp Eliminases

The seven top variants of the primary screening described above were purified and their Michaelis–Menten profiles of rate versus substrate concentration were determined at pH 7 (Figure 4A and Appendix A). This secondary screening confirmed the results of the primary screening, as all the selected variants showed substantially enhanced catalysis with respect to the library background (Figure 4A and Table 1). In particular, the variant with the sequence GLR at the three targeted positions showed an enhancement in catalytic parameters of approximately one order of magnitude over the library background. Note that most of the activity determinations reported in this work were performed at pH 7. However, it is known that the activity of Kemp eliminase enzymes based on the proton-abstraction mechanism may increase at basic pH, reflecting the deprotonation of the amino acid residue that gives rise to the catalytic base (the aspartate at position 229 in this case). Accordingly, we determined the profile of rate versus substrate concentration for the best GLR variant at pH 8.5. A significant increase in activity with respect to pH 7 was observed (Figure 2D), leading to a catalytic efficiency of k_cat_/K_M_~2·10^5^ M^−1^ s^−1^ and a turnover number of k_cat_~635 s^−1^ (Table 1).

It is interesting that the achieved improvements in catalysis did not come with a significant cost in stability, as shown by the denaturation temperature values determined via differential scanning calorimetry (Figure 4B,C and Table 1). The lack of a substantial activity-stability trade-off can be attributed to the fact that our engineering approach did not manipulate already-existing interactions, but introduced new ones.

### 2.4. Structural Analysis of the Catalysis Enhancement

The two best Kemp eliminases from the library screening shared the presence of a bulky hydrophobic residue at the second position after the carboxyl terminus. It appears reasonable to assume that this hydrophobic residue establishes interactions that stabilize the transition state of the reaction and enhance catalysis. In order to explore this possibility, we applied AlphaFold2 [39,40] to the prediction of the 3D structure of our best Kemp eliminases (Figure 5). We carried out the prediction for sequences in which the function-generating mutation (tryptophan to aspartate at position 229 in the β-lactamase sequence) was omitted (Note, however, that the overall 3D structure predicted by AlphaFold2 remained essentially the same upon the W229D mutation: see Appendix A). The reason for this is that W229 in β-lactamases is in roughly the same position as the bound transition state in the variants with the W229D mutation that showed Kemp elimination activity. Therefore, possible interactions that stabilize the bound transition state in the W229D variants may be suggested by the corresponding interactions with the tryptophan at position 229 in structures in which the W229D mutation is not included. Indeed, such interactions between the tryptophan and the hydrophobic residue at the second position are clearly suggested by the AlphaFold2 predictions (Figure 5). Finally, it is worth noting that the enhancement observed in the Kemp elimination activity was very unlikely to have been related to the lactam antibiotic activity of the β-lactamases we used as scaffolds for engineering, given that the two active sites (de novo for Kemp elimination and natural for antibiotic degradation) were well apart in both predicted and experimental structures (see Appendix A).

### 2.5. On the Possibility of Further Enhancements of Kemp Eliminase Activity

The catalysis levels achieved in this work were certainly substantial. However, it seems reasonable to ask whether they could be further enhanced using the same general approach. As we screened less than 10% of the different amino acid sequences spanned by the combinatorial library, it is possible that better variants could be found by additional screening of the same library. More relevant is the fact that the library we used was sharply focused on three positions, which allowed us to screen a substantial fraction of the library in a reasonable time. Nevertheless, it is conceivable that mutations at farther positions in the inserted polypeptide can also increase the rate of Kemp elimination. To explore this possibility, we prepared a new 8000-variant combinatorial library focused at the fourth, fifth, and sixth positions after the carboxyl terminus (Figure 3C). Screening of approximately 250 variants from this library yielded results that were qualitatively similar to those obtained with the first library. Most of the variants displayed significant activity in the primary screening (Figure 3C), perhaps reflecting that the library samples the sequence space associated with a conformationally flexible polypeptide segment that likely remained largely exposed to the solvent in most cases. However, as was the case with the first library, a few of the variants were found to display substantially enhanced levels of catalysis for the Kemp elimination reaction (Figure 3C), a result that was confirmed by variant protein purification and by determination of the Michaelis–Menten profiles and catalytic parameters (Figure 4A and Table 1). Certainly, the catalysis enhancement obtained on the basis of a limited screening of this second library was smaller than that afforded by the GLR variant from the first library. However, it is clear that combined screening of the positions included in the libraries is likely to lead to additional increases in de novo catalysis (work in progress).

## 3. Discussion

The generation of de novo enzymes (i.e., enzymes with completely new active sites) is one of the major unsolved problems in protein engineering [32]. In addition to the obvious biotechnological implications, the results of de novo enzyme engineering studies may have immediate implications for our understanding of the origin of life. Most of the chemical reactions of life are extremely slow in the absence of enzyme catalysis. Several analyses support the view that diverse and specialized enzymes were already present in the last universal common ancestor [41,42]. It would seem reasonable to assume that efficient molecular mechanisms for the de novo emergence of enzymes and their subsequent optimization must exist. However, such efficient mechanisms are not apparent in the efforts of protein engineers to develop completely new enzyme functionalities. Only a limited number of de novo enzymes have been reported to date [32], and several of the most recent success stories in this field involve the recruitment of metals or metal-containing cofactors that already provide, by themselves, some starting level of catalysis. However, only approximately 30% of enzymes are metalloenzymes [43,44], and the mechanisms for the emergence of new enzymes that do not rely on metal recruitment remain poorly understood. Furthermore, most of the engineered de novo enzymes display very low activities; many rounds of laboratory-directed evolution, a highly time-consuming procedure, are often required to bring their catalysis to levels that are similar to those of modern natural enzymes [37,45,46,47].

Starting with the work of Tawfik, Baker and coworkers in 2008 [48], Kemp elimination has been extensively applied as a benchmark of de novo enzyme engineering. The reaction can occur through a base-catalyzed mechanism, as shown in Figure 1A, and it is in fact considered as a model for proton abstraction from carbon, a fundamental process in chemistry and biochemistry. It can also occur through a redox mechanism. Highly active de novo enzymes based on the recruitment of the heme cofactor for Kemp elimination catalysis have been recently reported [49,50]. Here, however, we are concerned with “traditional” Kemp elimination that is achieved through proton abstraction by a catalytic base (Figure 1A).

The best base-promoted Kemp eliminase to date (at least in terms of k_cat_) was reported by Hilvert et al. in 2013 [36] and displays a catalytic efficiency of k_cat_/K_M_ = 2.3·10^5^ M^−1^·s^−1^ and a turnover of k_cat_ = 700 ± 60 s^−1^, values that compare favorably with those for the average modern natural enzyme (k_cat_/K_M_ about 10^5^ M^−1^·s^−1^ and a turnover of k_cat_~10 s^−1^; [35]). Remarkably, the catalytic parameters for the best Kemp eliminase found in this work, k_cat_/K_M_ = (2.0 ± 0.1)·10^5^ M^−1^·s^−1^ and k_cat_ = 635 ± 59 s^−1^, match those for the best Kemp eliminase as previously reported in the literature. However, these two efficient de novo Kemp eliminases are the result of quite different design approaches and screening efforts. Hilvert’s Kemp eliminase was the outcome of 17 rounds of directed evolution from a rationally designed de novo enzyme with a low but significant level of Kemp eliminase activity [36]. The starting background for this extensive screening effort was the result of a complex iterative procedure in which a failed (i.e., inactive) initial computational design was rescued on the basis of amino acid replacements that were suggested by analyses of molecular dynamics simulations and 3D crystallographic structures [51]. In contrast, the best Kemp eliminase reported here started with a minimalist design that targeted a conformationally flexible region in an ancestral β-lactamase scaffold [33]. Thus, a single mutation (W229D) generated a significant level of Kemp elimination activity that could be immediately enhanced by a second mutation (F290W) at the de novo active site. The Kemp elimination activity of this W229D/F2900W variant could be increased via computationally focused, ultra-low-throughput screening [34]. Specifically, screening of only 20 active-site variants at the top of the stability ranking predicted by the FuncLib approach [52] led to a substantial activity improvement. Finally, in this work we achieved further catalysis enhancement on the basis of the screening of only approximately 800 variants from a library that samples the sequence space of a short polypeptide segment engineered at the active site region.

Our success in arriving at an efficient base-promoted Kemp eliminase on the basis of a rather modest screening effort has immediate implications for the engineering of de novo enzymes. First, the catalysis-enhancing mutations identified in this work and in our previous work [33,34] do not increase the complexity of the catalytic machinery, which remains a simple proton abstraction by a catalytic base, as established by the function-generating mutation W229D. Rather, they appear to optimize interactions that stabilize the transition state for the reaction ([34,37] and this work). Overall, it emerges that large enhancements in de novo enzyme catalysis can be achieved through the fine-tuning of intramolecular interactions in the active site region. Second, this work demonstrated that a short polypeptide segment inserted near the new active site has the capability of generating such catalysis-enhancing interactions. This approach has several obvious advantages. The inserted polypeptide segment will, in principle, be flexible and exposed to the solvent to a substantial extent; thus, it will be unlikely to lead to disruptive interactions that compromise proper protein folding. In addition, being a short segment, the associated sequence space can be efficiently sampled on the basis of a moderate screening effort.

The proof of principle provided in this work takes advantage of the fact that the carboxyl terminus in β-lactamases is close to the location of the new active site; therefore, it can be easily used as the point of attachment of the new segment. This does not mean, however, that the extra-segment approach necessarily requires that the de novo active site is engineered near the carboxyl (or the amino) terminus of the protein. In fact, the new active site could be placed in any appropriate region of the protein (i.e., where design is successful), and then the carboxyl (or amino) terminus could be moved to a position close the new active site by engineering a protein variant with a suitable circular permutation [53].

## 4. Methods and Materials

### 4.1. Site-Saturation Mutagenic Libraries

Library preparation was performed using the QuikChange Lightning PCR method (Agilent # 210518). Three positions were simultaneously saturated with the mutagenic primers described in Appendix A in order to generate a combinatorial library comprising 8000 different amino acid sequences. The recombinant plasmid pET24-GNCA4-12-5G_HT containing the gene of the background variant was used as template. The amplification reaction contained 5 μL of 10× QuikChange Lightning Buffer, 1 μL of dNTP mix, 1.5 µL of QuikSolution reagent, 1.25 μL of primers (10 μM each mix), template plasmid (50 ng), 1 µL of QuikChange Lightning Enzyme, and water, to a final volume of 50 μL. The conditions for the PCR were as follows: 1 cycle at 95 °C for 2 min, 18 cycles of denaturation at 95 °C for 20 s, annealing at 60 °C for 10 s, and extension at 68 °C for 3.5 min. The final extension step was carried out at 68 °C for 5 min. The PCR products were digested with DpnI. Two μL of the commercial DpnI solution were added to the PCR sample and the solution was incubated at 37 °C for 5 min. Afterwards, 2 μL of this solution were used to transform *E. coli* XL10-Gold Ultracompetent cells (45 μL) with a heat pulse. Subsequently, cells were suspended in 1 mL of SOC medium, incubated for 1 h at 37 °C, and plated on LB-agar containing 100 μg/mL kanamycin. To test the quality of the library, ten clones were randomly selected, their plasmids were extracted, and the gene of the β-lactamase variants was sequenced (Sanger).

### 4.2. Library Screening

*E. coli* BL21 (DE3) cells were transformed with the plasmid containing the mutant libraries or, as a control the variant used as background for library construction (Figure 3A), plated on LB-Kan agar and grown for 16  h at 37 °C. Individual colonies were picked and transferred into 44 mm deep well plates containing LB-Kan medium (0.2 mL) using a Pickolo colony picker with a Freedom EVO 200 robot from TECAN (Männedorf, Schweiz). Each plate contained an internal standard with the variant used as library background (column 7, rows A to H) and a negative control (column 1, row H). These master plates were incubated at 37 °C with shaking at 250 rpm. After 16 h, clones from this pre-culture were inoculated (using a cryo-replicator CR1000 from Enzyscreen, Haarlem, The Netherlands) into deep well plates with fresh LB-Kan (lysate plates). After 4 hours of incubation at 37 °C, 250 rpm, LB-Kan with IPTG was added. The plates were incubated at 25 °C with shaking at 250 rpm for 16 h. The plates were centrifuged at 3000 g, the medium was discarded, and the cell pellets were frozen at −80 °C. After ~2 h the frozen cell pellets were re-suspended in HEPES 100 mM, pH 7. After 60 min at 25 °C the lysates were centrifuged at 3000× *g* and the supernatant was used for the Kemp eliminase assay.

### 4.3. Kemp Eliminase Assay for Library Screening

With the help of a liquid handler station (Freedom EVO 200, TECAN, Männedorf, Schweiz), 100 µL of the supernatant from lysate plates were transferred to the reaction plates. The initial activities and residual activities values were determined by adding 100 μL of HEPES 100 mM buffer pH 7.0 containing 0.25 mM 5-nitrobenzisoxazole. Plates were stirred briefly and the absorption at 380 nm (extinction coefficient of 15,800 M^–1^ cm^–1^) was followed in kinetic mode in the plate reader (Tecan M200 Infinite Pro Microplate Reader, Männedorf, Schweiz). The values were normalized against the average value corresponding to the background variant used for library construction. The best variants according to this determination were subsequently prepared and tested on pure form, as described below.

### 4.4. Protein Expression and Purification

The various β-lactamase variants studied in this work were prepared and purified as described previously [33,34]. Briefly, genes cloned into a pET24-b vector with resistance to kanamycin were transformed into *E. coli* BL21 (DE3) cells. The proteins were prepared with a His-tag and purified by affinity chromatography. Stock solutions for activity determinations and physicochemical characterization were prepared by exhaustive dialysis against the desired buffer.

### 4.5. Determination of Profiles of Rate versus Substrate Concentration for Kemp Eliminases

Rates of Kemp elimination were determined by following product formation by measuring the absorbance at 380 nm. An extinction coefficient of 15,800 M^−1^ cm^−1^ was used to calculate rates from the initial linear changes in absorbance with time. Most measurements were performed at 25 °C in HEPES 10 mM NaCl 100 mM pH 7 and 1% acetonitrile, although determinations in the same buffer at pH 8.5 were also performed for the best Kemp eliminase found in this work. As the stock solution of the substrate is prepared in acetonitrile, a certain concentration of this cosolvent in the reaction mixture is unavoidable. Note, however, that the concentration of acetonitrile stated (1%) was the final concentration and it was the same for all the experiments. That is, variable amounts of acetonitrile were added to solutions with different substrate concentrations to ensure constancy of the final acetonitrile after adding different volumes of the substrate stock solution in acetonitrile. In this way, it was guaranteed that no artefactual distortion of the curvature of the Michaelis plots arose because of variable acetonitrile concentrations. The absorbance increase at 380 nm was linear during the measurement and was recorded during a 15s interval, ensuring constant initial velocity conditions. All activity measurements were corrected by a blank performed under the same conditions. However, the determined rates were always clearly above the blanks. Profiles of rate versus substrate concentration were used to calculate the values of the catalytic efficiency (k_cat_/K_M_), the turnover number (k_cat_), and the Michaelis constant (K_M_) by fitting the Michaelis–Menten equation to the experimental data, as we described previously [33,34]. It is to be noted that the experimental substrate concentration range was limited by substrate solubility and that, for an acetonitrile concentration in the reaction mixture of 1%, substrate concentrations above 1 mM are not possible. Increasing the amount of acetonitrile in the reaction mixture certainly increases substrate solubility and the maximum accessible substrate concentration, but it also increases the value of the Michaelis constant. With an experimental substrate concentration range of 0–1 mM and a Michaelis constant on the order of a few mM, only a small curvature can be observed in the experimental Michaelis plots, a fact that could be thought to compromise the reliable determination of K_m_ and k_cat_. It is important to note, however, that, for each of the Kemp eliminases characterized in detail in this work, we obtained and analyzed three different profiles of rate versus substrate concentration, starting with at least two different protein preparations. Similar curvatures and congruent values of K_m_ and k_cat_ were observed from the analysis of the three independent Michaelis profiles that were determined for each variant (Figure 2C,D, Appendix A).

### 4.6. Protein Stability Determinations

Thermal stabilities of all the β-lactamase variants studied in this work were assessed by differential scanning calorimetry in HEPES 10 mM NaCl 100 mM pH 7 using a VP (Valerian Plotnikov) Capillary DSC (Microcal, Malvern), following protocols that were well established in our laboratory [7]. A typical calorimetric run involved several buffer–buffer baselines to ensure proper equilibration of the calorimeter, followed by runs with several protein variants with intervening buffer–buffer baselines. A single calorimetric transition was observed in all cases. We used the denaturation temperature, defined as the temperature corresponding to the maximum of the calorimetric transition, as a metric of protein stability.

### 4.7. Protein Structure Prediction

The protein structure prediction of variants obtained by directed evolution was performed using the ColabFold notebook implemented in Google Colab [40], based on the AlphaFold2 algorithm [39]. Sequences were used as inputs, and structure prediction was performed with default parameters. PyMOL (PyMOL Molecular Graphics System, Version 2.4.1 Schrödinger, LLC, created by Warren Lyford Delano, Delano Scientific LLC, San Francisco, CA, USA) was used to visualize the predicted models and to inspect the roles of the different mutations.

## Figures and Tables

**Figure 1 ijms-23-08934-f001:**
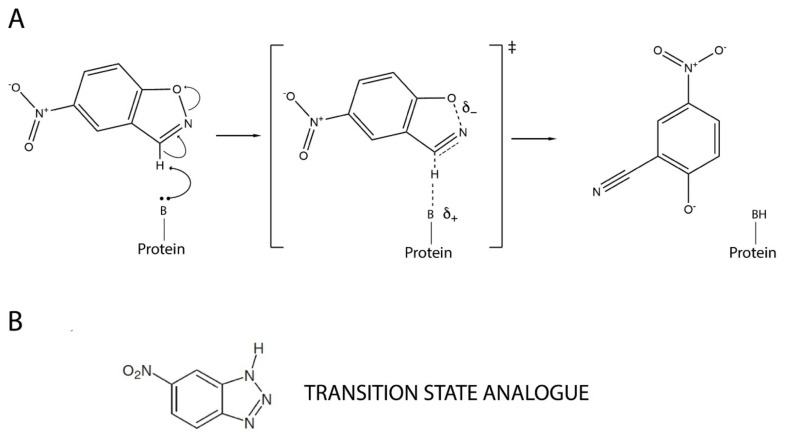
Mechanism of base-catalyzed Kemp elimination showing a proposed transition state structure (**A**). A transition state analogue, 5(6)-nitrobenzotriazole, is also shown (**B**).

**Figure 2 ijms-23-08934-f002:**
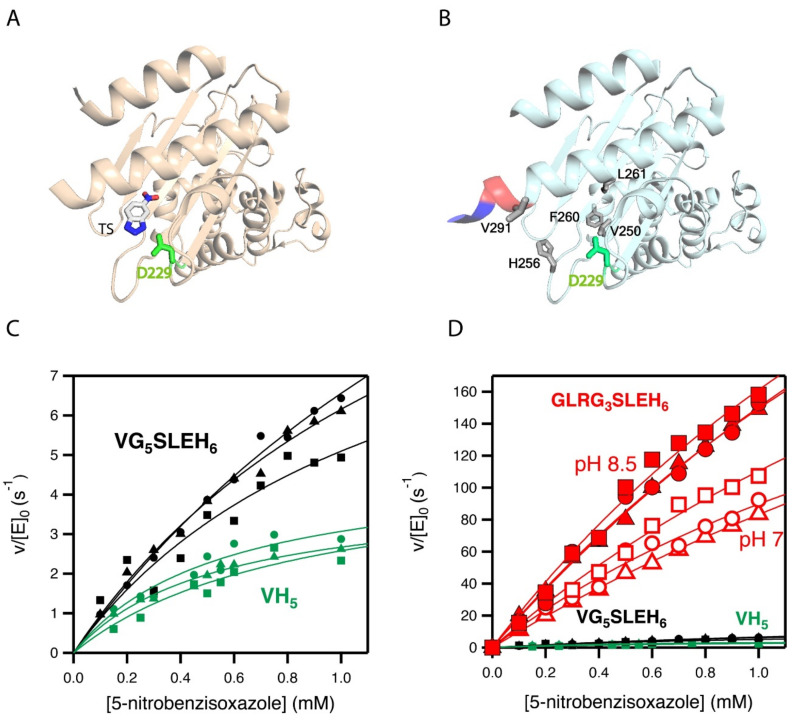
3D structures and catalytic properties of the Kemp eliminase variants used as starting points for the enzyme engineering reported in this work. (**A**) Structure (PDB ID 5FQK) of the W229D/F290W variant of an ancestral β-lactamase with Kemp eliminase activity [33]. The W229D mutation generates the new function and the catalytic base (D229) introduced by the function-generation mutation is shown. The structure includes a transition state analogue (see Figure 1), which indicates the location of the new active site. (**B**) The Kemp elimination activity of the protein shown in A could be enhanced by several amino acid replacements at the de novo active site region, resulting from computationally focused ultra-low-throughput screening [34]. The activity-enhancing residues are highlighted in grey in the structure shown here (PDB ID 6TXD). The His-tag attached to the carboxyl terminus for purification purposes is also highlighted (red and blue colors). Note that one of the activity-enhancing mutations replaces the first histidine with a valine. Therefore, the protein shown has a VH_5_ tail attached to the carboxyl terminus. The background variant used in this work has VG_5_SLEH_6_ attached to the carboxyl terminus, introducing a polypeptide segment between the caxboxyl terminus and the His-tag, but keeping the valine. (**C**) Michaelis–Menten profiles for the proteins with VH_5_ and VG_5_SLEH_6_ attached to the carboxyl terminus. (**D**) The differences in catalysis observed in C are very small compared with the activity enhancement achieved in the screening efforts reported this work (variant with GLRG_3_SLEH_6_ attached to the carboxyl terminus). Note that three Michaelis–Menten profiles were independently determined for each of the variants shown in (**C**,**D**).

**Figure 3 ijms-23-08934-f003:**
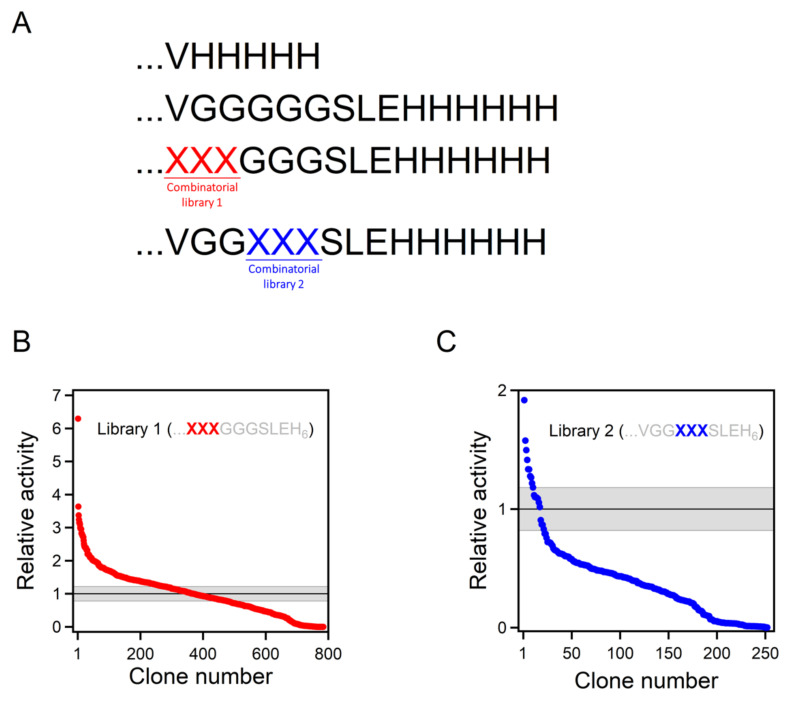
Combinatorial library screening for enhanced Kemp eliminase activity. (**A**) While the best Kemp eliminase from our previous work (34) had a VH_5_ tail attached, the background variant used for library construction in this work had a VG_5_SLEH_6_ polypeptide attached to the carboxyl terminus. Two 8000-variant combinatorial libraries were prepared, including all possible combinations of the 20 amino acids at two sets of three positions (labelled XXX), as shown. (**B**,**C**) The result of the screening of the two libraries. Clones are ranked according to Kemp eliminase activity relative to the background variant. The grey strip represents the average activity of the background variant plus or minus the associated standard error.

**Figure 4 ijms-23-08934-f004:**
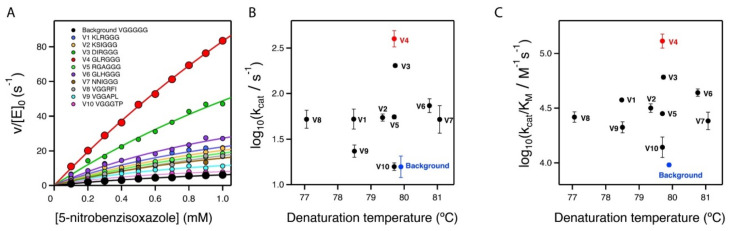
Secondary screening of the top variants from the primary library screening shown in Figure 3. (**A**) Michaelis–Menten profiles for the top variants at pH 7 (profiles for the best variant at pH 8.5 are shown in Figure 2B). The sequences at the relevant section of the included polypeptide are shown. Note that, for all variants, three independent Michaelis–Menten profiles were determined (see Appendix A); however, for the sake of clarity, only one representative profile for each variant is shown here. (**B**,**C**) Plots of Michaelis–Menten catalytic parameters for all the variants versus denaturation temperature, as determined by differential scanning calorimetry. These plots are scattergrams, indicating the absence of a significant stability/activity trade-off.

**Figure 5 ijms-23-08934-f005:**
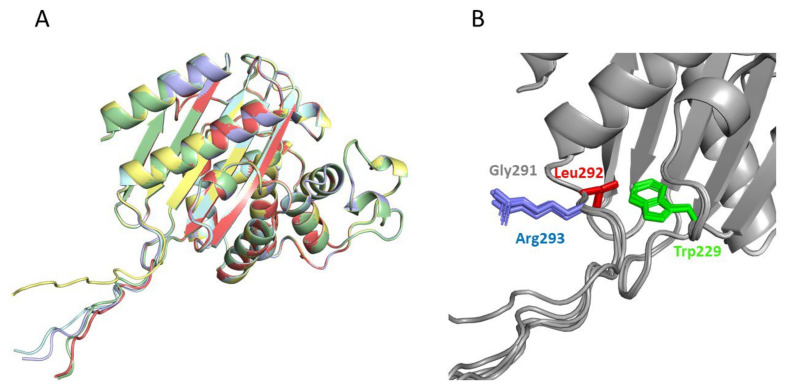
Prediction of the structure of the best Kemp eliminase obtained in this work by the program AlphaFold2 [39]. The prediction was carried out with a sequence that had a tryptophan at position 229, i.e., the function-generating W229D mutation was omitted. The reason for this was that the tryptophan at position 229 occupied a position close to that of the transition structure in the active Kemp eliminase (Figure 2A). Therefore, interactions with W229 in the predicted structure may conceivably correspond to interactions with the transition state that affects catalysis in the W229D variants. The top five predicted structures (**A**) are very close to the experimental structures shown in Figure 2, although the additional GLRG_3_SLEH_6_ polypeptide appears mostly extended and exposed to the solvent. However, a blow-up of the de novo active site region (**B**) reveals an interaction between the leucine of the polypeptide and the tryptophan at position 229 that could correspond to a kinetically relevant interaction with the transition state in the Kemp eliminase.

**Table 1 ijms-23-08934-t001:** Catalytic parameters for the cleavage of 5-nitrobenzisoxazole at pH 7 (HEPES 10 mM NaCl 100 mM) and 1% acetonitrile and 25 °C catalyzed by the engineered and evolved versions of Precambrian β-lactamases. For the best variant, data at pH 8.5 are included.

Variant	Sequence ^a^	k_cat_ (s^−1^) ^b^	K_M_ (mM) ^b^	k_cat_/K_M_ (s^−1^ M^−1^) ^b^	Tm (Cº) ^c^
V4 at pH 8.5	…GLRGGG…	635.0 ± 59.1	3.14 ± 0.49	(2.0 ± 0.1) × 10^5^	-
V4	…GLRGGG…	407.5 ± 76.7	3.35 ± 0.91	(1.3 ± 0.2) × 10^5^	79.7
V3	…DIRGGG…	202.9 ± 8.9	3.35 ± 0.21	(6.1 ± 0.3) × 10^4^	79.7
V6	…GLHGGG…	74.7 ± 13.5	1.74 ± 0.46	(4.4 ± 0.3) × 10^4^	80.8
V1	…KLRGGG…	54.1 ± 13.2	1.45 ± 0.37	(3.6 ± 0.1) × 10^4^	78.5
V2	…KSIGGG…	54.7 ± 4.8	1.75 ± 0.32	(3.2 ± 0.3) × 10^4^	79.4
V5	…RGAGGG…	55.4 ± 2.7	1.97 ± 0.1	(2.82 ± 0.01) × 10^4^	79.7
V8	…VGGRFI…	53.5 ± 11.0	2.10 ± 0.61	(2.6 ± 0.3) × 10^4^	77.0
V7	…NNIGGG…	55.4 ± 20.8	2.50 ± 1.43	(2.5 ± 0.4) × 10^4^	81.1
V9	…VGGAPL…	23.7 ± 3.8	1.12 ± 0.16	(2.1 ± 0.2) × 10^4^	78.5
V10	…VGGGTP…	15.9 ± 1.5	1.19 ± 0.32	(1.4 ± 0.3) × 10^4^	79.7
BACKGROUND	…VGGGGG…	16.3 ± 4.4	1.71 ± 0.48	(9.6 ± 0.4) × 10^3^	79.9

^a^ The sequences of each variant at the randomly mutagenized regions in the libraries are shown. Variants are ranked according to the catalytic efficiency. All values correspond to pH 7, with the exception of the best variant (V4), for which data at pH 7 and pH 8.5 are provided. ^b^ The values shown for the Michaelis–Menten catalytic parameters are the average values from three independent replicates (*n* = 3 independent determinations of the Michaelis–Menten profiles). The associated standard errors are also provided. See Appendix A for the results of the analysis of the individual profiles. ^c^ Denaturation temperatures determined by differential scanning calorimetry (DSC) are shown. The error associated with T_m_ determination by DSC is typically smaller than one degree.

## Data Availability

All relevant data are included in the manuscript and in the Appendix A.

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
