# Peer review of "Efficient Base-Catalyzed Kemp Elimination in an Engineered Ancestral Enzyme"

_ijms, 2022, doi:10.3390/ijms23168934_

Round 1

Reviewer 1 Report

General comment:

This manuscript, entitled “Efficient base-catalysed Kemp elimination in an engineered ancestral enzyme,” authored by Gutierrez-Rus et al., reported the strategic denovo protein engineering for the high catalytic activity or new function. This work describes some implications of designing new active sites based on the ancestral reconstruction approach, which may or may not be a true picture of the adaptability of the function. This is an excellent work for protein engineering and the basis for any biotechnological application. In my opinion, this is a valuable work and is suitable for publication in the International Journal of Molecular Sciences after the authors have addressed the following comments and questions:

Specific comments:

1)     Did the author find any allosteric effect by creating any mutation away from the active site?

2)     What is the basis for choosing neighboring or active site mutation? Is it just a mutation from the background version to further enhance activity? Same question about different versions like v1, v2, v3, v4…

3)     Please correct me if I am wrong -  whether 800 or 8000? Line 189   - Two 8000-variant combinatorial libraries were prepared, including all possible combinations of the 20 amino acids at two sets of positions,

4)     How challenging is it to perform an activity assay on a large scale? – is there variation if performed in replicates?

Author Response

We thank the reviewer for his/her supportive comments of our work. We have considered all the specific comments of the reviewer:

Reviewer: “Did the author find any allosteric effect by creating any mutation away from the active site?”

RESPONSE: This is certainly a very interesting possibility. However, predicting hot spots for long-distance effects in proteins is challenging (see, for instance, WIREs Comput. Mol. Sci. 11, e1502, 2021). In this work, therefore, we have not purposely explored the long-distance approach, inasmuch as the approach based on adding an extra polypeptide segment at the de novo active site was found to be successful.

Reviewer: “What is the basis for choosing neighboring or active site mutation? Is it just a mutation from the background version to further enhance activity? Same question about different versions like v1, v2, v3, v4…”

RESPONSE: The mutations were not really chosen. We simply sequenced the top variants from the primary screening (Sanger sequencing) and found the corresponding combinations of residues at the randomised positions that enhance activity with respect to the reference variant (which we call the background). This point is now clearly noted in the revised version (section 2.2).

Reviewer: “Please correct me if I am wrong -  whether 800 or 8000? Line 189   - Two 8000-variant combinatorial libraries were prepared, including all possible combinations of the 20 amino acids at two sets of positions,”

RESPONSE: It is 8000, because each of the two libraries randomizes 3 positions (20x20x20). I guess that, to avoid confusion, we should have written “…all possible combinations of the 20 amino acids at two sets of three positions”. This has been corrected in the revised version. We have also changed the terminology in figure 3 (using XXX to refer to the randomized positions) and provided additional information in panel A of figure 3, as well as in the figure legend, to make the library design clearer to the reader.

Reviewer: “How challenging is it to perform an activity assay on a large scale? – is there variation if performed in replicates?”

RESPONSE: Kemp elimination is easily followed by measuring absorbance as a function of time. Performing high-throughput activity analysis, therefore, would not be particularly challenging in terms of experimental design. However, it would have been extremely time consuming. All activity determinations (Michaelis-Menten profiles) reported in our manuscript are the result of three independent determinations performed with, at least, two different protein preparations. This is explicitly noted in the revised version (section 4.5).

Reviewer 2 Report

In the present study, the authors obtained a very active enzyme with no decrease in Tm value by simply creating mutant enzymes near the novel active site that they had previously created. This research is very significant in guiding the creation of enzymes with novel activity and is very worthy of publication.

I have some suggestions for improvement.

(1) It would be easier to understand what the authors have done in this study (adding further mutations near the newly created active site at the C-terminus) and the reason for it if the authors added more information in the abstract and introduction. It is a little difficult to know from the abstract and introduction what methods and experiments were used to obtain such highly active mutant enzymes.

(2) Line 63-75

Where is the original active site of the ancestral Precambrian beta-lactamase with broad substrate specificity?

It was somewhat confusing to the less knowledgeable reader that the lactamase activity of Precambrian beta-lactamase is not related to the activity of kemp elimination.

(3) L241-250

The indole ring when position 229 is Trp is similar to the transition state analog, but does the predicted structure of the highly active mutant enzyme (GLRG3SLEH6) overlap with that of the complex?

Is there little information that can be obtained from the superimposition of the complex structure and the mutant enzyme?

What would be the structure of this C-terminal flexible loop if a model of the mutant containing W229D were made with AlphaFold2? Does AlphaFold2 predict the structure where the loop is far from the active site?

(4) Fig.2, Table 1, Fig. S1, and Fig. S2

The Km values are doing a little worse, although they are overshadowed by the dramatic increase in the kcat values of the mutant enzymes. Do the authors have any interpretation on this matter?

The concentration of the substrates (0-1 mM) in the activity assay is below the Km values. Although the value of kcat/Km can be calculated from the linear part at low substrate concentrations, it is not possible to accurately estimate the values of kcat and Km. Why do the authors measure only up to 1 mM when it should be better to measure activity up to at least twice the Km value? It is inevitable that the substrate is not soluble in water, it would be better to indicate kcat and Km as apparent values.

Author Response

We thank the reviewer for his/her positive comments and his/her useful suggestions for improvement. We have considered all of them:

Reviewer: “(1) It would be easier to understand what the authors have done in this study (adding further mutations near the newly created active site at the C-terminus) and the reason for it if the authors added more information in the abstract and introduction. It is a little difficult to know from the abstract and introduction what methods and experiments were used to obtain such highly active mutant enzymes.”

RESPONSE: We have followed the reviewer’s suggestion and have included in the abstract and the introduction sections additional information about the approach used.

Reviewer: “(2) Line 63-75

Where is the original active site of the ancestral Precambrian beta-lactamase with broad substrate specificity?

It was somewhat confusing to the less knowledgeable reader that the lactamase activity of Precambrian beta-lactamase is not related to the activity of kemp elimination.”

RESPONSE: We have included in the Supplementary Information of the revised version an additional figure (Supplementary Figure 2) in which the two active sites (the Kemp elimination active site and the antibiotic degradation active site) are indicated. It is apparent that they are clearly distinct and spatially separated active sites. This point is emphasized in the revised version (section 2.4)

Reviewer: “(3) L241-250

The indole ring when position 229 is Trp is similar to the transition state analog, but does the predicted structure of the highly active mutant enzyme (GLRG3SLEH6) overlap with that of the complex?

RESPONSE: Yes, it does. We have carried out the Alphafold2 prediction for the highly active variant and they agree well with the structure of the complex from a previous study, except, of course, for the presence of the extra segment in the highly active variant from this work. In the revised version, the superposition of the two structures is shown in panel C of the new Supplementary Figure 2 and the agreement is noted in the main text (section 2.4).

Reviewer: “Is there little information that can be obtained from the superimposition of the complex structure and the mutant enzyme?”

RESPONSE: We think that no critical new information is to be gained from the superposition, since the two structures agree very well with each other.

Reviewer: “What would be the structure of this C-terminal flexible loop if a model of the mutant containing W229D were made with AlphaFold2? Does AlphaFold2 predict the structure where the loop is far from the active site?”

RESPONSE: Yes, the AlphaFold2 prediction is that the extra segment extends away, as shown in the new Supplmentary Figure 2. Still, this prediction is likely kinked to the fact that the extra segment is not natural and AlphaFold2 does not have sequence correlation information about the segment. It is likely that the segment is flexible and actually exist as an ensemble of different conformations, many of which will be more or less extended, although some other conformations may involve some interaction with the de novo active site.

Reviewer: “4) Fig.2, Table 1, Fig. S1, and Fig. S2

The Km values are doing a little worse, although they are overshadowed by the dramatic increase in the kcat values of the mutant enzymes. Do the authors have any interpretation on this matter?

RESPONSE: The Km values fluctuate somewhat among the different variants, but they remain in the mM range. It is to be noted that Km was also in the mM range for the very active Kemp eliminase reported by Hilvert and coworkers in Nature in 2013 (ref. 36). Overall, we feel that the changes we see in Km (much less than one order of magnitude) are too small to warrant a detailed discussion on their origin, at least at this stage.

Reviewer: “The concentration of the substrates (0-1 mM) in the activity assay is below the Km values. Although the value of kcat/Km can be calculated from the linear part at low substrate concentrations, it is not possible to accurately estimate the values of kcat and Km. Why do the authors measure only up to 1 mM when it should be better to measure activity up to at least twice the Km value?

RESPONSE: The substrate concentration range in our experiments is determined by solubility. It is not possible to go beyond 1 mM substrate concentration in 1% acetonitrile. Of course, it is possible to increase solubility of the substrate by using, say, 5% acetonitrile in the solution. However, nothing is gained by doing this, because the Km value increases with acetonitrile concentration. We mention this briefly in the revised version (section 4.5).

Reviewer: “It is inevitable that the substrate is not soluble in water, it would be better to indicate kcat and Km as apparent values.”

RESPONSE: The fact that the substrate range is limited by solubility, does not imply that the Kcat and Km values determined from the experimental Michaelis profiles are apparent. Rather, it only increases the error associated to the determined values. However, for each variant and condition, we have performed three independent determinations with, at least, two different protein preparations. Furthermore, in all cases, curvature (which carries the information about the individual values of Km and kcat) is clearly observed in the Michaelis plots. Therefore, the values given for kcat and Km are reliable within the reported errors derived from the three independent determinations. These points are discussed in some detail in the revised version (section 4.5).